# An Evaluation of the Quality of Plain Radiograph Interpretations by Radiology Trainees: A Single Institution Experience

**DOI:** 10.3390/diagnostics12081954

**Published:** 2022-08-12

**Authors:** Nurliyana Izyan A Halim, Faizah Mohd Zaki, Hanani Abdul Manan, Zahiah Mohamed

**Affiliations:** 1Department of Radiology, Universiti Kebangsaan Malaysia Medical Center, Jalan Yaacob Latif, Bandar Tun Razak, Cheras, Kuala Lumpur 56000, Malaysia; 2Department of Radiology and Intervention, Hospital Pakar Kanak-Kanak (Children Specialist Hospital), Universiti Kebangsaan Malaysia, Jalan Yaacob Latif, Bandar Tun Razak, Cheras, Kuala Lumpur 56000, Malaysia; 3Makmal Pemprosesan Imej Kefungsian, Department of Radiology, Universiti Kebangsaan Malaysia Medical Center, Jalan Yaacob Latif, Bandar Tun Razak, Cheras, Kuala Lumpur 56000, Malaysia

**Keywords:** diagnostic radiology, medical education, training, plain radiographs, radiology trainee

## Abstract

Introduction: The primary communication between the radiologist and referrer is through the radiological report. However, there are incidents of misinterpretation during radiologist training. Therefore, the present study aimed to evaluate the accuracy level and incidence of interpretation errors for plain radiographs among radiology trainees at our institution. Materials and Methods: The present study retrospectively reviewed 508 reported plain radiographs for one year, and two radiologists subsequently evaluated these plain radiographs. The initial diagnosis by the trainee was compared with the radiologists’ evaluation, and the results were categorized as either ‘accurate’, ‘minor discrepancy’, or ‘major discrepancy’. The data were analyzed concerning the overall performance, year of trainee, anatomic area, patient age group, and radiograph type. A chi-square test was performed, with *p* < 0.05 indicating statistical significance. Results: The overall accuracy rate was 69%, with minor and major discrepancy rates of 21% and 10%, respectively. There was an insignificant increase in overall accuracy with increased years of training, despite a reduction to 58% accuracy among Year 3 trainees. The accuracy level increased between Year 1, Year 2 and Year 4 by 70%, 71% and 75%, respectively (*p* > 0.05). The accuracy rates for both the adult and pediatric age groups were not statistically significant. The mobile radiographs showed lower accuracy rate of reporting than the plain radiographs. Conclusion: The radiological trainee interpretations for plain radiographs had an average rating with low discrepancy rates. The Year 3 trainees had the lowest accuracy compared to the other trainee groups. However, the present study suggests the need for further research to determine if the current outcomes are outliers or are indicative of a real phenomenon.

## 1. Introduction

Plain radiography remains the most common form of radiological imaging performed in any medical institution despite advanced radiological modalities. Advantages of plain radiography include its low radiation dose, availability, affordability, and simplicity [1]. The plain radiograph has a lower sensitivity rate than the computed tomography (C.T.) scan, thus a greater degree of interpretation is required [1]. It is well known that the primary communication between the radiologist and the referrer is through the radiological report [2], which is considered a medico-legal document that contains the official interpretation of a single radiological examination or procedure [2]. A sufficient report must be accurate, concise, and clearly understood by the referring team. This is vital, as the radiological report may affect the primary team’s subsequent management and the patient’s outcome. Our radiology department is part of a tertiary teaching university hospital that has provided Master of Radiology residency training for over 20 years. Furthermore, our 800-bed facility performs approximately 70,000 plain radiographs annually. A critical 24 h deadline for trainee submission of radiographic interpretations has been streamlined into the daily workflow to facilitate student training and aid our primary team’s ability to provide better patient care. Nevertheless, radiology trainees come from different radiological backgrounds and experiences, and instances of missed findings occasionally result in amended reports. When junior trainees experience difficulty interpreting radiographs, common practice at our institution has been to seek senior opinion.

With an increased volume of radiographic reading and reporting, trainees gain valuable experience in detecting abnormalities. This is also the case with other modalities, such as ultrasound, fluoroscopy, computed tomography, and magnetic resonance imaging (MRI). The trainee’s initial interpretation provides the basis for the patient’s immediate management, especially when on-call or after office hours [2,3]. Every radiologist is concerned about giving a false-positive reading or missing a diagnosis in their daily practice [3]. Lee et al., reported a retrospective error rate of approximately 30% for radiological interpretation. In addition, roughly 4% of radiologic interpretations adjudicated by radiologists in daily practice contain errors, and nearly 75% of medical malpractice claims against radiologists are related to diagnostic errors [3]. According to Wallis et al., radiologist trainees commonly fail to model the practice of reasonable radiograph interpretation [2].

Previously, there has been no formal study conducted at our institution to assess trainee interpretations of plain radiographs. In order to maintain high-quality practice within our radiology department, performance must be systematically monitored, analyzed, and improved [4,5]. Therefore, this retrospective, cross-sectional study aimed to assess the quality of plain radiograph interpretation among radiology trainees. Our hypothesis was that increased consecutive years of training would result in increased accuracy of the trainees’ plain radiograph interpretation.

## 2. Materials and Method

The hospital ethics (Institutional Review Board) committee approved the present cross-sectional study. The patients’ consent was waived due to the retrospective nature of this study. Five hundred eight plain radiograph reports were randomly selected from the radiology information system (RIS) for one year. Twenty-five radiograph reports were selected every second and fourth Monday of the month, alternating between general and mobile radiography. Unreported radiographs and irretrievable images were excluded.

The radiograph images were reviewed via Picture Archiving, Communication System, Digital Imaging, and the Communications in Medicine systems. The radiographs were interpreted and reported in a local reporting system called the Integrated Radiology Information System (IRIS^®^). A list of patients, along with the dates and times of their plain radiographs, was given to two consultant radiologists for blinded review. They were instructed to indicate their findings and primary diagnosis for each patient. Any discrepancies between the two radiologists’ reports were deliberated upon until a consensus was reached.

The initial diagnosis by the trainee and the final diagnosis by both radiologists were recorded and categorized. The categorizations were designated with the following labels: ‘Accurate’ meant that the radiologists agreed with the report, and ‘Mild discrepancy’ meant that there was a difference in interpretation that otherwise, did not determine the patient’s healthcare. Meanwhile, ‘Major discrepancy’ meant that the difference in interpretation would likely have had a significant impact on the patient’s management.

The information was then compiled based on the following: date, request number, patient name, medical registration number (MRN), patient age and date of birth, anatomic area, clinical data, verification of trainee’s report, term year of trainee, radiologists’ reports, and final report categorization. The patient’s data were kept confidential. Statistical analysis was performed using the chi-square (*X*^2^) test to calculate the correlation between the accuracy and discrepancy rates, and *p* < 0.05 was considered statistically significant. 

## 3. Results

### 3.1. Number of Radiographs by Year of Trainees

This study included 508 radiographs taken from a 1-year period. First-year trainees reported 122 radiographs (24%); second-year trainees reported 96 radiographs (19%); third-year trainees reported 113 radiographs (22%); fourth-year trainees reported 177 radiographs (35%).

### 3.2. Overall Performance

The radiologists agreed with 69% (*n* = 351) of the reports made by the trainees. The remaining 31% of reports contained discrepancies, with minor discrepancies in 21% of cases (*n* = 107) and major discrepancies in 10% of cases (*n* = 50). 

### 3.3. Overall Performance by Year of Trainees

Generally, the level of accuracy increased progressively with years of training, with Year 1 at 70% (*n* = 85), Year 2 at 71% (*n* = 68), and Year 4 at 75% (*n* = 132). Unfortunately, there was reduced accuracy in Year 3 relative to the other groups, with 58% (*n* = 66). Hence, this led to an insignificant statistical value of *p* = 0.958 (Table 1).

In conjunction with the above findings, there was a reduction in total discrepancies (both minor and major discrepancies) from Year 1 to Year 4, as Year 1 had 30% (*n* = 37), Year 2 had 29% (*n* = 28), and Year 4 had 26% (*n* = 45). Results indicate that Year 3 had the highest level of total discrepancies, at 42% (*n* = 47). The variance level of minor discrepancies did not progressively decrease, with 22% (*n* = 27) among Year 1, 18% (*n* = 17) among Year 2, 28% (*n* = 32) among Year 3, and 18% (*n* = 31) among Year 4, for an average of 21.5% (*p* value = 0.917) between the four groups (Table 1). Major discrepancies increased between Year 1 and Year 3, ranging from 8% to 11%, but then dropped to 8% for Year 4 (*p* value = 0.953) (Table 1). In short, the statistical value for all accuracy variance (accurate, minor, and major discrepancies) was insignificant, as *p* > 0.05 (Table 1). 

### 3.4. Accuracy Variance Based on Anatomic Area

The largest portion of radiographs performed were images of the chest, accounting for 56.5% (*n* = 287). Spine and knee radiographs constituted 9.3% (*n* = 47) and 5.3% (*n* = 27), respectively. The rest of the anatomic areas accounted for the remaining 5%. The orthopantomogram (OPG) was the least-performed radiograph, as it represented only 0.2% (*n* = 1).

When major discrepancies were classified by anatomic area, the kidney, ureter, and bladder (KUB) radiograph had the highest incidence level, at 25%. Foot, abdomen, and skull radiographs demonstrated the second-worst value, at 20.3%. Major discrepancies were low for shoulder, wrist, chest, ankle, and spine radiographs, ranging from 17% to 4% in descending order. The trainees made no statistically significant major discrepancies for the humerus, elbow, radius/ulna, hand, knee, femur, tibia/fibula, pelvis, or OPG radiographs. 

### 3.5. Overall Accuracy by Age Group of Patients

The present study included 445 total radiographs (87.5%) for the adult age group and 63 radiographs (12.5%) for the pediatric age group. The accuracy levels for the two age groups were not significantly different, at 70% for the adult age group and 63% for the pediatric age group. Minor discrepancies made by trainees were less common among the adult age group than for the pediatric age group, at 9% and 17%, respectively.

The major discrepancies made by trainees for both age groups were not significantly different, with an average of 20%.

### 3.6. Overall Accuracy by Type of Radiographs

In this study, there were a total of 329 general radiographs (64.8%) and 179 mobile radiographs (35.2%). The accuracy rate was higher for general radiographs, at 75% (*n* = 248), than it was for mobile radiographs, at 58% (*n* = 103). The discrepancy rate for general radiographs was lower than for mobile radiographs, at 25% (*n* = 81) and 42% (*n* = 73), respectively. Some examples of major discrepancies in the mobile group consisted of missing the following diagnoses: pneumothorax, pneumomediastinum, pulmonary venous hypertension, and pneumonia. Examples of such cases are demonstrated in Figure 1, Figure 2, Figure 3 and Figure 4.

## 4. Discussion

In the present study, we found that the accuracy level of plain radiograph interpretations among trainees at our facility low in comparison to other reports. The present study discussed the possible reason for these findings and our plan to mitigate them in the future. 

Previous studies have reported a 97.37% accuracy level in their trainees’ radiograph interpretations [6] and indicated that the major discrepancy rates in the study were less than 1% [5,6,7]. In another study, Ruchman et al., reported that the interpretation discrepancy rates between radiology residents and attending radiologists involved all modalities, and that the overall major discrepancy rate was 2.6% [4]. These reports suggest that our accuracy level is lower than that of other institutions. There are a few factors that may contribute to this lower accuracy level. One potential explanation would the difference in sample sizes, as the sample size in the selected studies is larger than ours. Previous studies have interpreted a wider variety of cases than ours and included CT scan interpretations while our study did not [4,6]. 

Numerous studies have shown increasing accuracy rates across successive years of training [6,7]. Nevertheless, at our institution, there was a decline in performance among Year 3 trainees, as they had the lowest level of accuracy and highest level of discrepancies. Such findings are consistent with previous studies by Ruchman et al., and Wysoki et al., as both found that the discrepancy rate was highest for third-year residents [4,8]. Branstetter et al., found that radiologists, residents, and junior attendees had lower error rates than seniors, though the differences were not statistically significant [5]. We propose that this phenomenon could be attributed to the transition from being in the junior pool (Year 1 and Year 2) to being in the senior (Year 3 and Year 4), which may have led to uncertainty and stress on account of the increased responsibility. Additionally, they may have lower confidence and heightened emotional instability due to the approaching primary examination at the end of the semester.

Nevertheless, as the level of training advances, the difficulty of the radiographs also increases. Whereas junior trainees typically seek advice from seniors when they are uncertain about their findings, seniors are supposed to consult with attending radiologists when uncertain about their findings. It is hoped that trainees will come to develop a sense of professionalism, and will learn to be aware of their limitations and when to ask for help. 

Our study showed that the chest radiograph is the most frequently performed radiological investigation. There were a total of 287 chest radiographs included in the present study, and they were interpreted with 63% accuracy (*n* = 182), 26% minor discrepancy (*n* = 73), and 11% major discrepancy (*n* = 32). The accuracy of trainee chest radiograph reporting was low compared to other studies, and we discovered that the major discrepancies were generally due to missed diagnoses. The majority of missed diagnoses involved pneumonia and congestive cardiac failure. Infrequently, they involved lung mass, pneumothorax, pneumomediastinum, respiratory distress syndrome, and transient tachypnea in newborns. A study by Grasvenor et al., found 93% agreement between radiologists and trainees, but 11% (of 200) reports led to immediate changes in patient management [9]. Another author’s study described a discrepancy rate of 1% (134/12,600) between trainees and radiologists in identifying the presence and absence of pneumonia from chest radiographs [10]. 

The KUB radiograph had the highest rate of major discrepancy in the present study, due to a missed diagnosis of a ureteric calculus. Turk et al., reported that KUB radiographs have a sensitivity of 44–77% and a specificity of 80–87% to identify calculus [11]. Acute renal colic due to ureteric calculus obstruction is a medical emergency that requires immediate pain management. A low dose CT has become the preferred method for detecting ureteric calculus [11]. The calculus being missed in KUB radiographs could be due to poor bowel preparation before the procedure, inadequate assessment along the course of the ureter, and no history being available during the reporting. 

Two studies disclosed that emergency department clinicians had significantly higher error rates in radiograph interpretations than radiological staff [5,12]. The trainees at our institution did relatively well in reporting trauma cases, with approximately 77% accuracy (108 out of 139 radiographs). The significant discrepancy rates were for foot, skull, shoulder, wrist, and ankle radiographs at 21%, 20%, 17%, 14% and 9%, respectively. These major discrepancy rates were predominantly instances of missed fractures, similar to previous findings [6,13]. The remaining anatomic areas (bone radiographs) involved trauma cases with a negligible degree of major discrepancy. 

The levels of accuracy and major discrepancy in radiograph interpretations by our trainees for both the adult and pediatric age groups were not significantly different. However, the minor discrepancy levels were higher for the pediatric age group, wherein the majority were due to over-reporting a normal chest radiograph. In the major discrepancies for the pediatric age group, however, the trainees tended to miss the diagnosis in the chest and abdominal radiographs, such as in cases of pneumothorax, transient tachypnea of newborn, respiratory distress syndrome, and abnormal positioning of an umbilical arterial or venous catheter. A study by Elemraid et al., on pediatric (children < 16 years old) chest radiograph interpretations found a significant inter-observer variability of 22%, for which the most frequent sources of disagreement were in the reporting of patchy and perihilar changes [14]. Several similar studies also found this tendency in diagnosing bacterial pneumonia; alveolar infiltrates and pleural effusion are findings with high intra- and inter-rater reliability [15,16,17].

Mobile radiographs are mainly performed for bedridden patients, with mobile chest radiographs being the most common. The image quality for portable radiographs is significantly lower than that general radiographs, due to the projection of the radiograph, (which is usually done in anteroposterior view), unsatisfactory positioning of the patient, the image being taken when the patient is under inspiration, and poor patient exposure. One study revealed that the overall accuracy of chest radiographs in diagnosing pneumonia in bedridden patients was 69%, sensitivity and specificity were 65% and 93%, respectively, and positive and negative predictive values were 83% and 65%, respectively [18]. These findings are concurrent with our study, which showed a 58% accuracy rate for mobile radiographs. 

The present research has several limitations that could have affected our findings. The first limitation is found inherently in the nature of a retrospective study, in which there are delays in both the interpretation of results and identification of trends. During our research, trainees of varying levels reported the radiographs randomly, and junior trainees only sought help from senior trainees when they encountered difficulty interpreting the radiographs. At present, mobile radiograph reports are only allowed by senior trainees who are given the responsibility to verify all the radiographs interpreted by a first-year trainee. The quality of radiographs is significantly increased when junior and senior residents are paired. Ref. [6] Multiple readings of radiographs, especially if complex, leads to higher sensitivity and lower error rates [10].

The present study’s second limitation was its relatively small sample size, with only 508 radiographs being included. Previous studies by Branstetter et al., Cooper et al., and Ruutiainen et al., had 1499, 93,132, and 33,024 cases, respectively (Table 2); the cases consisted of radiography, ultrasound, CT and MRI [5,6,7]. The relatively small sample size may contribute to the comparatively high level of discrepancy rates. 

In order to reduce major discrepancies, we have now summoned our radiologists to verify preliminary radiographs reported by the residents, and we encouraged them to give feedback when major discrepancies are encountered. Following the study, this has become our practice and has resulted in remarkable improvement in residency training, and the residents have indicated that they value the feedback. Although recent literature has discussed that artificial intelligence is comparable with the preliminary report of radiology resident in interpreting radiographs [19], we still believe training in plain radiograph interpretation is crucial to residency programs, and that it should still be a core component of the radiology residency program syllabi. 

## 5. Conclusions

In the present study, we found that the accuracy level of the plain radiograph interpretations among trainees in our facility is lower than that in other facilities. Several factors could be contributing to this lower accuracy. To avoid this issue in the future, we have since implemented new procedures requiring radiologists to verify the preliminary radiograph reports of the residents.

## Figures and Tables

**Figure 1 diagnostics-12-01954-f001:**
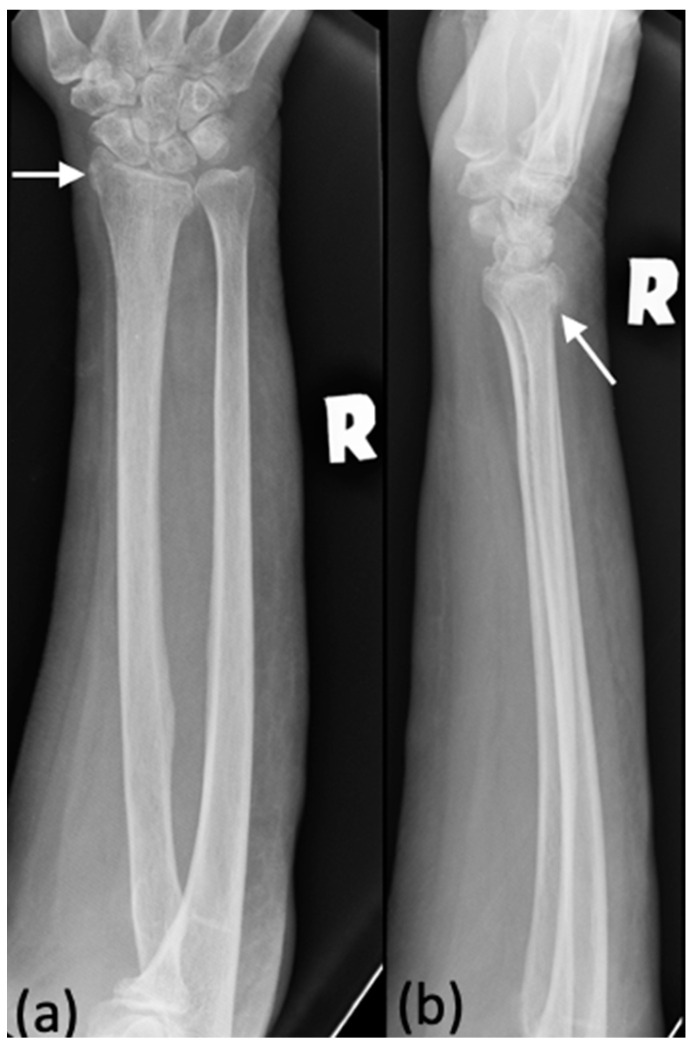
(**a**,**b**) An example of a radiograph categorized as a minor discrepancy. This right forearm radiograph was performed on a 25-year-old male who presented with pain after a sports injury. Both the trainee and radiologists agreed that this patient had a fracture of the distal end of his radius. However, the trainee did not report the angulation and dislocation, shown in (**b**) (white arrow).

**Figure 2 diagnostics-12-01954-f002:**
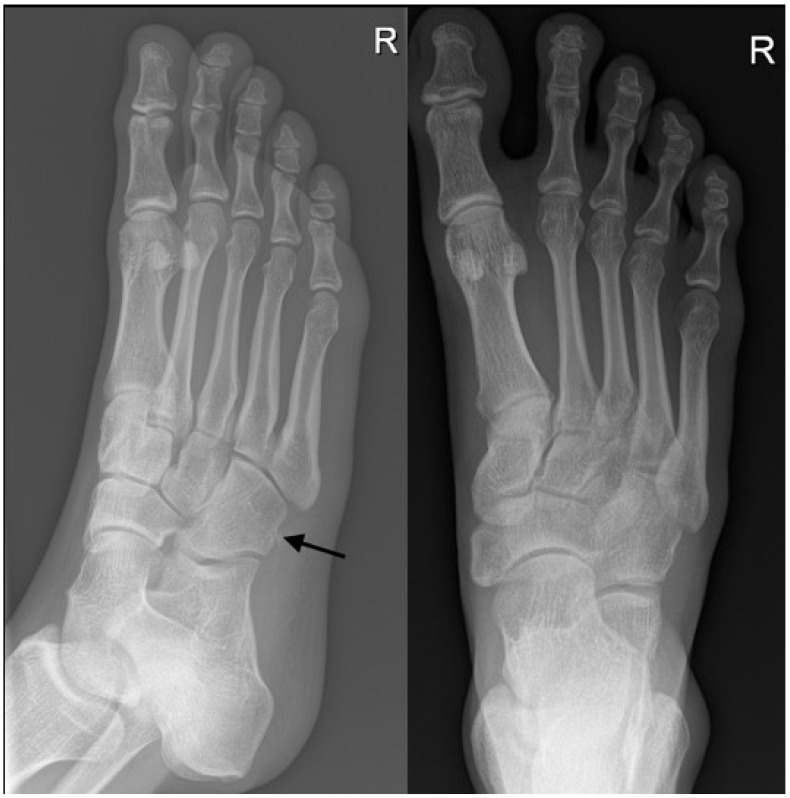
An example of a radiograph that was categorized as accurate. A foot radiograph was performed on a 56-year-old male with a history of falls. Both the trainee and radiologists agreed on there being cuboid fracture with chip bone fragment (black arrow).

**Figure 3 diagnostics-12-01954-f003:**
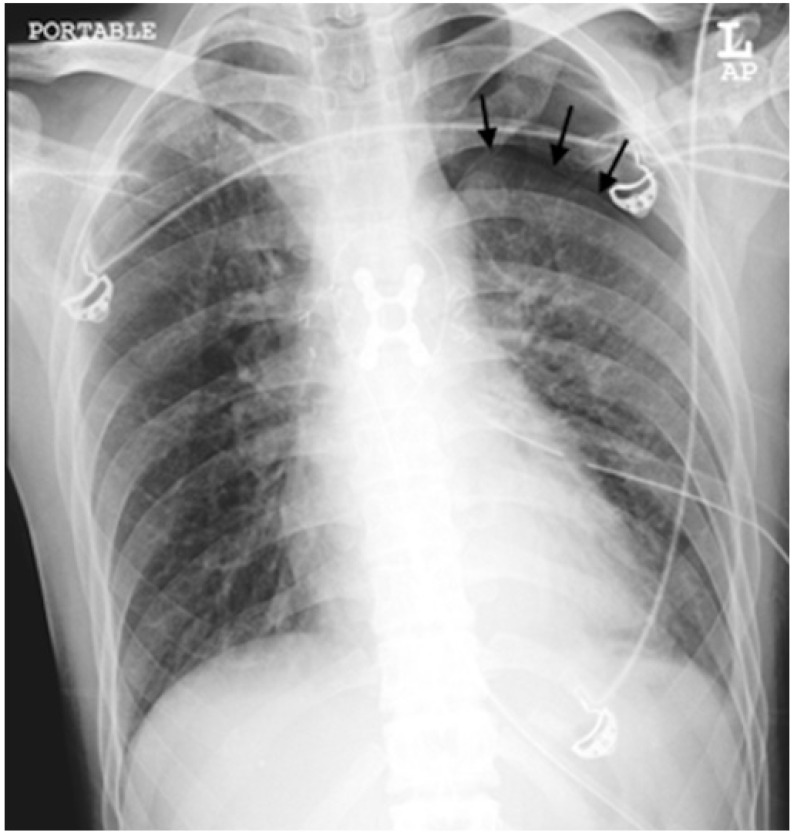
An example of a radiograph categorized as a major discrepancy. A chest radiograph was performed on a 50-year-old man with underlying bronchial asthma who presented with shortness of breath. The trainee missed the pneumothorax in the left hemithorax in about 20% of them (black arrows).

**Figure 4 diagnostics-12-01954-f004:**
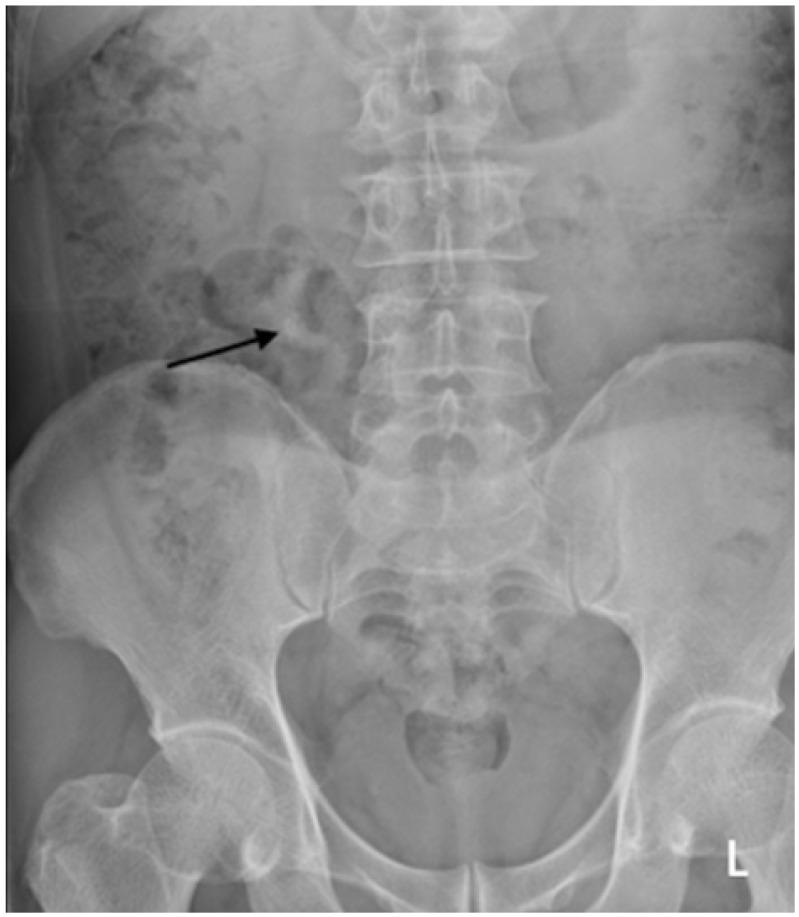
Another example of radiograph categorized as a major discrepancy. A kidney, ureter, and bladder (KUB) radiograph was performed on a 60-year-old man who presented with an acute abdomen. There is an opacity at the course of the right mid ureter with provisional right mid ureteric calculus (black arrow), which the trainee missed.

**Table 1 diagnostics-12-01954-t001:** Accuracy variance by year of trainees.

	Year 1, *n* = 122	Year 2, *n* = 96	Year 3, *n* = 113	Year 4, *n* = 177	*p* Value
Accurate, *n* (%)	85 (70)	68 (71)	66 (58)	132 (75)	0.958
Minor discrepancy, *n* (%)	27 (22)	17 (18)	32 (28)	31 (18)	0.917
Major discrepancy, *n* (%)	10 (8)	11 (11)	15 (13)	14 (8)	0.953

**Table 2 diagnostics-12-01954-t002:** A comparison of previous studies from 2007–2011.

	*n*	Accurate	Minor Discrepancy	Major Discrepancy
Branstetter et al. [5]	1499			0.8%
Cooper et al. [6]	93,132	97.37%	1.98%	0.59%
Ruutiainen et al. [7]	33,024			0.89%
Our current study	500	69%	21%	10%

## Data Availability

The data presented in this study are available on request from the corresponding author. The data are not publicly available due to institution policy.

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
