# Peer review of "An Evaluation of the Quality of Plain Radiograph Interpretations by Radiology Trainees: A Single Institution Experience"

_diagnostics, 2022, doi:10.3390/diagnostics12081954_

Round 1
Reviewer 1 Report
A small study looking at accuracy of residents reading xrays.
Results are repeated extensively throughout the Discussion. This needs to be rectified by rewriting this portion of the paper. Do not repeat Results in the Discussion.
Discussion - There should be some comment about why the R3's performance was so poor. Was it lack of supervision, poor quality of resident, lack of training, poor instructions to this cohort? There needs to be some comment.
- Why was there such poor quality with these Malaysian residents? There should be some comment when others have shown a rate of 1% of inaccuracy. Why so high a discrepancy? Certainly not ascceptable and this study seems to accept it as ok? Why?
- KUB paragraph is not really pertinent to study as there is little explanation about why this program had such a high rate of misreading KUBs. Why did this happen? Lack of training? Lack of supervision? few hours spent reading?
- How about some numbers about your program explaining how many hours spent reading each year/ how much time spent in supervised reading/ how much time in unsupervised reading/ how to rectify misdiagnoses/ program changes based upon study?
- Conclusion - Generally too long. Get to the point. First sentence is stating a result from the Result section again. Rewrite.
- second sentence is possibly wrong
- what changes are to be made in the program because of the study?
Author Response
|
No. |
Comments/Suggestions
|
Response |
|
|
Reviewer 1 A small study looking at accuracy of residents reading xrays. Results are repeated extensively throughout the Discussion. This needs to be rectified by rewriting this portion of the paper. Do not repeat Results in the Discussion. Discussion - There should be some comment about why the R3's performance was so poor. Was it lack of supervision, poor quality of resident, lack of training, poor instructions to this cohort? There needs to be some comment. Why was there such poor quality with these Malaysian residents? There should be some comment when others have shown a rate of 1% of inaccuracy. Why so high a discrepancy? Certainly not ascceptable and this study seems to accept it as ok? Why? KUB paragraph is not really pertinent to study as there is little explanation about why this program had such a high rate of misreading KUBs. Why did this happen? Lack of training? Lack of supervision? few hours spent reading? How about some numbers about your program explaining how many hours spent reading each year/ how much time spent in supervised reading/ how much time in unsupervised reading/ how to rectify misdiagnoses/ program changes based upon study? Conclusion - Generally too long. Get to the point. First sentence is stating a result from the Result section again. Rewrite. second sentence is possibly wrong what changes are to be made in the program because of the study?
|
Amendments and corrections have been done accordingly.
Amendments and corrections have been done accordingly. In page 8.
Amendments and corrections have been done accordingly.
Amendments and corrections have been done accordingly
Amendments and corrections have been done accordingly
Amendments and corrections have been done accordingly |

Reviewer 2 Report
The proposed study aimed to assess the quality of plain radiograph interpretation among radiology trainees. There are some major and minor issues to be improved.
That said, I believe this manuscript could be suitable for the general audience of Diagnostics.
Major Revision
To evaluate the consistency of the obtained results, I suggest you to perform an inter-rater agreement/correlation (e.g. Fleiss'Kappa), which measures the reliability of different sets of measures by different raters
Minor revisions
Caption Figure 1 - Please change figure 3B (white arrow) with Figure 1B (white arrow)
Caption Figure 2 - Please change (white arrow) with (black arrow)
Author Response
|
Reviewer #2:
The proposed study aimed to assess the quality of plain radiograph interpretation among radiology trainees. There are some major and minor issues to be improved.
That said, I believe this manuscript could be suitable for the general audience of Diagnostics.
Major Revision
To evaluate the consistency of the obtained results, I suggest you to perform an inter-rater agreement/correlation (e.g. Fleiss'Kappa), which measures the reliability of different sets of measures by different raters
Minor revisions
Caption Figure 1 - Please change figure 3B (white arrow) with Figure 1B (white arrow)
Caption Figure 2 - Please change (white arrow) with (black arrow)
|
We are very sorry, and we are unable to do the test. No available radiologist to evaluate images.
Amendments and corrections have been done accordingly
|
Round 2
Reviewer 1 Report
Good revision - no deficiencies noted
Author Response
Dear Reviewers and Editor,
Diagnostics. MDPI.
Thank you very much for your suggestions and comments. You have helped us see the manuscript differently and tremendously improved the manuscript.
Hanani Abdul Manan
